# Characteristics of Households’ Vulnerability to Extreme Heat: An Analytical Cross-Sectional Study from India

**DOI:** 10.3390/ijerph192215334

**Published:** 2022-11-20

**Authors:** Lipika Nanda, Soham Chakraborty, Saswat Kishore Mishra, Ambarish Dutta, Suresh Kumar Rathi

**Affiliations:** 1Public Health Foundation of India, Gurugram 122002, India; 2Indian Institute of Public Health, Bhubaneswar 751013, India; 3Centre for Management Studies, Administrative Staff College of India, Hyderabad 500034, India; 4Department of Central Research and Innovation Center, Sumandeep Vidyapeeth Deemed to be University, Vadodara 391760, India

**Keywords:** extreme heat, vulnerability, climate, urban, India

## Abstract

High ambient temperature is a key public health problem, as it is linked to high heat-related morbidity and mortality. We intended to recognize the characteristics connected to heat vulnerability and the coping practices among Indian urbanites of Angul and Kolkata. In 2020, a cross-sectional design was applied to 500 households (HHs) each in Angul and Kolkata. Information was gathered on various characteristics including sociodemographics, household, exposure, sensitivity, and coping practices regarding heat and summer heat illness history, and these characteristics led to the computation of a heat vulnerability index (HVI). Bivariate and multivariable logistic regression analyses were used with HVI as the outcome variable to identify the determinants of high vulnerability to heat. The results show that some common and some different factors are responsible for determining the heat vulnerability of a household across different cities. For Angul, the factors that influence vulnerability are a greater number of rooms in houses, the use of cooling methods such as air conditioning, having comorbid conditions, the gender of the household head, and distance from nearby a primary health centre (PHC). For Kolkata, the factors are unemployment, income, the number of rooms, sleeping patterns, avoidance of nonvegetarian food, sources of water, comorbidities, and distance from a PHC. The study shows that every city has a different set of variables that influences vulnerability, and each factor should be considered in design plans to mitigate vulnerability to extreme heat.

## 1. Introduction

High ambient heat and heatwaves are two of the major sources of weather-related mortality worldwide. Several studies have pointed out that a rise in temperature leads to an increase in all-cause mortality [1,2,3]. Climate is changing very rapidly and in an uneven manner; however, climate models are less likely to accurately predict the severity and duration of heatwaves. Leaving aside all other potential factors, extreme heat-related illnesses can be easily linked to an increase in excess all-cause mortality. Rathi et al. (2021) revealed a 39 per cent increase in all-cause mortality when temperatures reach 45 °C and above, while another Indian study from Surat revealed that all-cause mortality increases by 60 per cent in a few regions of a city during dangerous heat periods [1,4]. Similar to high ambient heat, heatwave periods have also been known to directly impact human lives and livelihoods [5]. Heatwaves have killed more people since 2000 than avalanches and exposure to cold; cyclones; tornadoes; and famine as a result of natural disasters, earthquakes, epidemics, floods, landslides, heavy rain, and forest fires [6]. High extreme heat leads to the activation of physiological stressors that are the key ingredients for heat-related illnesses. Although mild heat-related illnesses can be reversible, severe heat-related illnesses are potentially fatal.

A heatwave or an extreme heat-related event affects at-risk populations. Heatwaves are also a social justice issue and a matter of public policy, as the poor often live in the hottest areas, including slums [7,8]. According to studies, heatwave susceptibility is the result of a combination of factors, including socioeconomic, physiological, and climatological characteristics [5,9]. All of these variables lead to either a rise in vulnerability or a rise in resilience.

Vulnerability is the state of a person, a household, or any system that is a result of a combination of numerous factors. Exposure, sensitivity, and adaptive ability are common classifications for these factors. The vulnerability of a household to heat is determined by its exposure, sensitivity, and adaptation capacity [10]. The intensity and spatial distribution of elevated temperatures [11] that elevate heat conditions are referred to as exposure. Sensitivity refers to a household’s ability to cope with increasing exposure or the degree to which greater exposure will physically harm the household [11]. The ability of a household to actively minimise or adjust to personal exposure [10,12], using available skills and resources to ensure survival and sustainability, is referred to as adaptive capacity [13]. The link between adaptive capacity and vulnerability has been described in three ways in previous research [14,15]. To begin with, vulnerability and adaptability are not mutually exclusive concepts. Second, susceptibility is caused by a lack of adaptive capability, as well as a variety of other factors. Finally, they are inversely proportionate, implying that greater capacity implies low vulnerability and vice versa. While an increase in vulnerability is caused by increased exposure and sensitivity, an increase in adaptive capacity reduces the inclusive risk [16]. According to Wilhelmi’s concept, a household’s heat vulnerability is a function of its exposure, sensitivity, and adaptive capacity [16]. Meanwhile, there is also the concept of resilience, which should also be considered when discussing the impact of extreme heat or related events. Resilience can be defined as individuals and communities exposed to calamities and emergencies having the ability to foresee, prepare for, limit the impact of, cope with, and recover from the consequences of shocks and pressures without jeopardising their long-term views [17]. This paper aims to identify the characteristics of households’ vulnerability to extreme heat and their coping practices for Indian urbanites in Angul and Kolkata.

## 2. Materials and Methods

### 2.1. Study Sites

The cities were chosen based on the recommendations of the Task Force on Heat Waves of the National Disaster Management Authority (NDMA), India; their topographical location; the Indian Meteorological Department’s (IMD) Heat Wave criterion (temperature ≥ 45 °C); and their representation of hilly (Angul) and coastal (Kolkata) areas. Angul is a large coalmining district, whereas Kolkata is a coastal city and a massive metropolis, with vastly diverse socioeconomic capacities. Kolkata has a population of around 4.4 million residents, while Angul has a population of around 44 thousand residents. Angul is hilly, landlocked, and is mainly an industrial city with several largescale industrial projects surrounding the small residential location. Kolkata, on the other hand, is a much more diverse city with a very high population density (22,000/km^2^ as compared to ~200/km^2^ for Angul). Kolkata is bordered by the Hooghly River (a tributary of Ganga) and is very near the coastal zone. Summers (March–June) are hot and humid for Kolkata, and during dry spells, maximum temperatures sometimes exceed 40 °C in May and June, while maximum temperatures exceed 45 °C from April to June for Angul.

### 2.2. Study Design

An analytical cross-sectional design is followed to achieve the objectives. The survey was conducted from April to October 2020.

### 2.3. Sample Size and Sampling Design

A total of 1000 household respondents (500 from Angul and 500 from Kolkata) from 18 to 60 years old were interviewed. A two-stage sampling design was used for the selection of the respondents. In the first stage, a total of 23 and 25 administrative wards were selected through a simple random sampling methodology for Angul and Kolkata, respectively. In the second stage, 20–25 households were selected randomly from the administrative wards.

### 2.4. Data Collection and Management for Household Survey

The questionnaire for vulnerability assessment was designed after considering a careful review of the literature [16,18,19,20], keeping in mind each city’s context. The questionnaire considered several domains with diverse varieties of inquiries including multiple choice, yes or no, and open-ended questions to obtain higher granularity in the data. Before being used in the field, the survey questionnaire was developed in English, translated into two local languages—Oriya for Angul and Bangla for Kolkata—and rigorously tested. Self-identified heads of households were given questions and response options in their native languages by field officers. Since it was felt that women were the most familiar with their family’s circumstances, they were favoured. Males were not disallowed, though, if they were the only ones there or if their wives decided not to reply. The average time to finish a survey was 40 min. After each field workday, an author reviewed all surveys to ensure they were accurate and thorough.

Assessment of Vulnerability Characteristics: The majority of survey questions were closed-ended and looked at different areas to understand the range of factors that could raise population vulnerability throughout the chosen cities. Socioeconomics, WASH, waste management, food and nutrition, housing, locational characteristics, community, risk perception, coping mechanisms, early warning, quality of life, comorbidities, habits, and livelihood/occupation were the domains taken into account. Individual and household-level survey questions were included; respondents acted as proxies for the members of their households.

The questionnaire was pre-tested on five per cent (25 households from each city) for determining the sequencing of the questions, the ease of understanding the questions, and the overall flow of the questionnaire and revised accordingly. These pilot-tested households were not part of the actual sampled population. Household data were collected by trained interviewers. Before taking their written consent, the participants were provided with a detailed participant information sheet.

### 2.5. Approach and Measurement

The heat vulnerability index (HVI) development’s rationale, methods, and approaches have been described elsewhere [21]. In brief, exposure, sensitivity, and adaptive capacity are influenced by a variety of individual household indicators that can affect heat vulnerability. All these indicators were detailed in an empirical study by Rathi et al. (2021), and evidence was also considered from other studies [5,21,22,23]. A multidimensional vulnerability index was created with the use of these indicators and computed through a simple average of the component indices for the dimensions, as used by Rathi et al. (2021) and Wolf et al. (2014) [21,22]. To group the households into low and high vulnerability, the mean HVI for each city was used as the cut-off point. Low-vulnerability households had HVI values less than the city’s average HVI, whereas high-vulnerability households had HVI values greater than the city’s average HVI. After the construction of the multidimensional index, bi-variate and multivariable analyses were performed with HVI as the outcome variable.

#### 2.5.1. Model Specification for Angul and Kolkata

The dependent variable in the model is the likelihood that a household is ‘highly’ vulnerable to extreme heat (vulnerability). It is assumed that a particular household’s vulnerability to extreme heat in Angul and Kolkata is affected by socioeconomic, demographic, ecological, and health factors. For Angul, the gender of the household head and the distance of the nearest primary health centre (PHC) from the place of residence are used to control for the social characteristics, while the economic aspect is measured by the number of rooms in the house (room). On the other hand, the household size is used to capture the demographic characteristics. The perceived change in temperature and use of air coolers or air conditioners (ACs) at the workplace are used as proxies for ecological factors. Finally, the mild symptoms of high ambient heat and comorbid conditions in the household members (*comorbid*) are expected to capture health conditions. Accordingly, the following functional relationship is envisaged for Angul:(1)Vulnerability=f(Gender,Distance,Room,Size,Temperature,Cooling,Symptoms,Co−morbid)

For Kolkata, the occupation of the household head and the distance of the nearest primary healthcare centre from the place of residence are used to control for the social characteristics, and the economic aspect is measured by the mean income of the household for the summer months and the number of rooms in the house. On the other hand, the total number of accessible sources of water (*water*) is used to capture the ecological aspect. The mild symptoms of high ambient heat (*symptoms*) and comorbid conditions in the household members (*comorbid*) are expected to capture health conditions. Finally, places for sleeping during hot nights (*sleeping*), the intake of nonvegetarian foods (*nonveg*), and changing the amount of food consumed (*food*) during extremely hot days are used as proxies for behavioural changes.

Accordingly, the following functional relationship is predicted for Kolkata:(2)Vulnerability=f(Occupation,Distance,Income,Room,Water,Symptoms,Co−morbid,Sleeping,Non−Veg food)

Vulnerability is measured in both cities’ models using a dummy variable that takes the value ‘1’ when a household’s HVI score is greater than 0.5 and ‘0’ otherwise. Table 1 provides descriptions of the independent factors as well as their likely impact on the outcome variable.

#### 2.5.2. Estimation Technique

In this study, the outcome variable (household vulnerability) is binary in nature, i.e., low vulnerability and high vulnerability. The low-vulnerability households are defined as those households that are in a vulnerable state but are still capable of managing without external assistance, while highly vulnerable households are in need of urgent assistance and can be resuscitated only with immediate, best-possible interventions. There are other competing models that determine factors affecting outcome/dependent variables. However, the goal of the current analysis is to explore the socioeconomic, demographic, and ecological characteristics that may change a household from a state of low vulnerability to a state of high vulnerability and vice versa, which may help in making interventions timely. Hence, the study uses a logistic regression model with a binary dependent variable to study the association between the binary response probability and independent variables. The dependent variable in a Logit model is the natural logarithm of the odd ratio, which is treated as a linear function of the explanatory variables, i.e.,
(3)$$L_{i}=\ln\left(\frac{P_{i}}{1-P_{i}}\right)=\alpha +\overset{k}{\underset{j=1}{\sum }}\beta_{j}X_{ij}+u_{i}$$

Here, P_*i*_ represents the number of independent variables included in the model.

The dependent variable’s likelihood of being true (i.e., taking a value of 1) follows a logistic distribution, i.e.,
(4)$$P_{i}=\frac{1}{1+e^{-z_{i}}}=\frac{e^{z_{i}}}{1+e^{z_{i}}}$$
where the statistics for the Logit model can be obtained because the current study used cross-sectional household-level data. As the dependent variable is binary, the Logit is the following: If the dependent variable is true and if the dependent variable is false. The Logit model is calculated using the maximum likelihood method of estimation.

In addition, the study applied both Pearson’s and the Hosmer–Lemeshow tests for goodness-of-fit for estimated Logit model.

## 3. Results

Characteristics of households’ vulnerability to extreme heat for Angul and Kolkata:

For Angul, we observed that the share of females witnessing high heat vulnerability (78.2%) is substantially greater than the share of males who are experiencing high heat vulnerability (64.9%). It is also credibly evident that as the distance of the nearest PHC from residences becomes greater, a larger share of households suffers from high heat vulnerability. For instance, 59.2% of the households who travel between 1 and 5 km to reach the nearest PHC experience high heat vulnerability. This figure is as high as 92.6% for those who are made to commute more than 5 km. Similarly, as a household’s perception of the extent of changes in the ambient temperature and humidity at home increases, a larger share of households is found to be suffering from high heat vulnerability. Moreover, a considerable share of households (94.4%) that use air-conditioners/air-coolers at the workplace is seen to have low heat vulnerability. On the contrary, a large share of households (87.9 %) that do not use air-conditioners/air-coolers at the workplace is seen to have high heat vulnerability. Further, shares of households with the presence of mild symptoms or comorbid conditions are seen to have a low vulnerability to extreme heat as compared with those without any such medical conditions (Table 2).

For Kolkata, the share of respondents witnessing high vulnerability is substantially higher for those working as clerks (83.0%) followed by those engaged in unskilled (68.3%), skilled/semi-skilled (65.9%), professional/semi-professional (65.9%), and self-employed/business (59.3%) activities. On the contrary, a large share (62.5%) of respondents working in the agriculture and related sectors had a low vulnerability to extreme heat. Furthermore, heat vulnerability is also higher for a large share (72.6%) of respondents who are unemployed. Interestingly, we observed that as the distance between the place of residence and the nearest PHC increases, a larger share (77.7%) of households’ experiences lower vulnerability. Further, considerable shares (42.9 and 43.5%) of households with the presence of mild symptoms or comorbid conditions are seen to have a lower vulnerability to extreme heat as compared with those who are without these medical conditions. A large share of households who predominantly sleep on mattress floors (80.0%) has a high vulnerability to extreme heat, followed by those who sleep on a bed (71.7%). On the contrary, the majority of households who sleep on bare floors (55.3%) experience low vulnerability to extreme heat. Moreover, households that have avoided the intake of nonvegetarian food or reduced the quantity of food consumption during summers face low vulnerability to extreme heat as compared with those who did not (Table 2).

The value of the likelihood ratio (χ2 (9) = 326.8) with a *p*-value of 0.001 for Angul and likelihood ratio (χ2 (17) = 123.2) with a *p*-value of 0.001 for Kolkata shows that these models, as a whole, fit significantly for these cities. Furthermore, the value of the Pseudo R2 is fairly high, signifying goodness of fit for the projected model.

The test statistics of Pearson’s χ2 tests and the Hosmer–Lemeshow χ2 tests indicate that the estimated Logit model does not have a goodness-of-fit problem. In the Logit model for Angul city, the coefficients for gender, distance (5 km and more), room, temperature, cooling, symptoms, and comorbid factors are statistically significant. The coefficients of gender, distance (5 km or more), and temperature are positive, but these are negative for the room, cooling, symptoms, and comorbid factors. This indicates that the female respondents have a higher vulnerability to extreme heat. Heat vulnerability is also higher for households that reside more than five kilometres away from the nearest PHC as compared with those that stay within a radius of one kilometre. Further, the households that have perceived a drastic increase in temperature and humidity in the last few years have a higher vulnerability to extreme heat.

On the other hand, vulnerability to extreme heat is lower for households that stay in houses with a greater number of rooms or use air-coolers/air-conditioners at their workplaces. It is also lower for people who have experienced mild symptoms of high ambient heat (such as headaches, dizziness, weakness, and muscle pain) during the summer. Vulnerability to extreme heat is also lower for households that have members with comorbid conditions, such as diabetes and hypertension.

However, the coefficients of size and distance (>1 km and <5 km) are not statistically significant. This means that a household’s vulnerability to extreme heat does not vary significantly, as it is dependent on the household size or if the distance of the house from the nearest PHC is within one to five kilometres (Table 3).

In the Logit model for Kolkata, the coefficients for occupation (unemployed, agricultural, and related), distance (5 km and more), income, room, water, symptoms, comorbid, sleeping (bare floor), and nonvegetarian factors are found to be statistically significant. While the coefficients of occupation (unemployed) and income are positive, they are negative for occupation (agriculture and allied), distance (5 km or more), room, water, symptoms, comorbid, sleeping (bare floor), and nonvegetarian factors. This means that, as compared with professional and semi-professional workers, heat vulnerability is greater for those who are unemployed. Interestingly, heat vulnerability is also higher for respondents with a higher average income during the summer months.

On the other hand, vulnerability to extreme heat is lower for respondents involved in agriculture and related activities as compared with those working as professionals or semi-professionals. Unlike what is observed for Angul city, heat vulnerability is lower for respondents who reside at a distance of more than five kilometres from the nearest PHC as compared with those who stay within a radius of one kilometre. Vulnerability to extreme heat is also lower for respondents who stay in homes with a greater number of rooms or access to more sources of water. In line with what is seen in the case of Angul, heat vulnerability is lower for respondents with mild symptoms of high ambient heat during the summer or comorbid diseases such as diabetes and hypertension. While it is true that heat conditions can alter human behaviour, subtle behavioural changes in humans can also dampen the adverse impacts of heat conditions. The results show that respondents who sleep on the bare floor (as compared with a bed) during hot nights or those who avoid the consumption of nonvegetarian foods during hot summers are more likely to have lower heat vulnerability (Table 3).

However, the coefficients of food, distance (>1 km and <5 km), and occupation (clerical, skilled/semi-skilled, unskilled, self-employed/business) are less likely to be significant. This reflects that respondents’ vulnerability to extreme heat does not vary significantly in relation to changes in food consumption quantity or if the distance between their house and the nearest PHC is within one to five kilometres. Heat vulnerability also does not significantly differ for respondents whether they work as clerks, skilled/semi-skilled workers, unskilled labour, or run their own businesses.

## 4. Discussion

The characteristics of household vulnerability to extreme heat in the cities of Angul and Kolkata in India’s eastern belt were examined in this paper. The Logit regression results find that heatwaves and extreme heat are more likely to affect women compared with men. The finding is similar to numerous studies that have shown that women disproportionately suffer the impacts of heatwaves [25,26], disasters [29], and climate change [30,31] in developing countries, especially in India. This could be related to unequal power dynamics, inequitable cultural and societal standards [29], and physiological or financial mechanisms. Women usually stay put at home doing household chores while men go out and work. However, higher vulnerability to extreme heat is possible when women lack basic sanitation facilities or lack adequate access to electricity, running water, and toilets or stay in rooms with limited air circulation at home. Indoor cooking and the use of unclean/dirty forms of fuel at home can also put women at a greater risk of heat extremes, as established by Balmes et al. [32]. In Angul, the majority of households continue to use coal for cooking purposes because coal is abundantly available to the local people, either freely or at significantly low prices [33]. The easy availability of coal is a major reason for the non-adoption of liquefied petroleum gas (LPG) or stacking up coal with LPG in Angul. Women (and children) suffer most from this because of spending more time indoors and essentially carrying out all of the cooking, especially in India [23,34]. As per a report by the Health Effects Institute (2020), nearly 600,000 people died in India due to indoor air pollution in the year 2019. Further, high LPG cylinder costs and the severe effect of the COVID-19 pandemic on the income of households made the use of LPG unaffordable for many.

Moreover, descriptive statistics indicate that a very small share of women uses protective gear (such as umbrellas, hats, headcovers, etc.) to prevent direct sunlight during extreme heat conditions compared with men. In addition, women face cultural restrictions on wearing clothing that is suitable for extreme heat. Traditional clothing makes it harder for women to cope with extreme heat compared with men. This possibly explains why women are more vulnerable to extreme heat compared with their male counterparts in Angul.

Extreme heat and heatwaves are more likely to affect households living more than 5 km from the nearest PHC. The Angul–Talcher region is one of India’s ten most highly polluted locations according to the Comprehensive Environmental Pollution Index (CEPI). Thus, households residing in close proximity to coal mines, thermal power plants, and traffic junctions are more likely to be affected by a range of airborne-emission-led health hazards. Such households are compelled to visit healthcare centres more frequently. Hence, both the higher frequency of visits and the long distance from the public healthcare centres increase the exposure to extreme heat, thereby making such households more vulnerable [35].

Lack of continuous electricity supply, the inadequate possession of durable electric appliances (such as fans, air-coolers, etc.), and stuffy houses may cause people to be more vulnerable to extreme heat. The residents of Angul have also witnessed unscheduled power cuts and low voltage during peak summers. This could have further exacerbated the existing indoor temperature and humidity, thereby raising vulnerability to extreme heat.

Heat vulnerability is lower for households who stay in houses with a greater number of rooms or use air-coolers/air-conditioners at the workplace [5]. A higher number of rooms may lead to lower heat vulnerability, especially when the size of the household is low. The use of air-coolers or air-conditioners at the workplace reduces the sensitivity of people to extreme heat conditions. Hence, such households are more likely to have relatively low heat vulnerability. Households residing in homes that have ceilings made of heat-trapping materials (such as tin sheets, cement, plastic, and tarpaulin) may also experience a drastic increase in indoor temperature and humidity. Such houses are more prevalent in urban slums.

People who experience mild symptoms of high ambient heat are more likely to remain at their residences in order to recuperate. Therefore, they are less vulnerable to extreme heat. Heat vulnerability is lower for households that have members with comorbid diseases such as diabetes or hypertension. It is likely that the household members who suffer from diabetes or hypertension belong to a higher age group and are, supposedly, not the primary bread earners. Diabetes and hypertension are frequent in India’s older and middle-aged groups, affecting people from all walks of life [36]. This relationship is observed to be even stronger in an urban setting as opposed to the rural areas of the country. This perhaps compels the rest of the family to provide constant care for them, resulting in reducing their outdoor activities. Hence, exposure to heat vulnerability is lower.

The Kolkata model finds that unemployed people are more prone to being affected by excessive heat and heatwaves compared with professional employees. Unemployed (or semi-employed) people are likely to go outside in search of work and a regular source of income. Eventually, the households with unemployed members end up spending the bulk of their days outside, thereby increasing their exposure to extremely high temperatures. It is also well established that unemployed people with lower incomes are a key factor that increases household vulnerability [37]. Such people usually walk through or use bicycles as a means of transport, thereby raising their exposure to extreme heat during the summer months. On the other hand, people involved in agricultural and related activities are likely to have lower heat vulnerability. Farmers in Kolkata are mostly engaged in single-harvest crops (either Rabi or Kharif) and remain relatively unengaged for half of the year. This may lead to low exposure and, hence, lower vulnerability to extreme heat. More so, people associated with farming and related activities are also more likely to develop better physiological adaptations because of their lifelong exposure to extreme heat conditions.

High-income people are generally found to be less vulnerable and more resilient to extreme heat because of their better coping capacity. In the case of Kolkata, this finding presents a paradox. In order to earn more income, the respective household members need to either pay more for outside visits or work overtime or diversify their economic activities. In either of these cases, the probability of becoming more vulnerable to extreme heat increases on account of high exposure.

Heat vulnerability is lower for respondents who reside at a distance of more than 5 km from the nearest PHC. This is contrary to what is observed for Angul. The frequency of medical visits by the respondents in Kolkata may be far less due to the lower burden of diseases compared with Angul. Qualitative analysis revealed that a considerable share of households in Kolkata prefer treatment at private healthcare facilities compared with Angul. A preference for seeking private healthcare treatment by households in Kolkata also stems from the fact that the average household income level in the city is relatively much higher than in Angul. Further, the mode and quality of transportation to and from the PHCs may be relatively better, with Kolkata having air-conditioned cabs, buses, and metro–railway networks to travel within the city with ease. These few factors together may result in lower heat vulnerability.

Like the experience in Angul, heat vulnerability in Kolkata is lower for respondents who stay in houses with a greater number of rooms. More rooms may lead to lower heat vulnerability. Moreover, having a higher number of rooms can facilitate separate areas for the kitchen, where the ambient heat is generally high.

Households that have multiple sources of water for domestic purposes are likely to have a lower vulnerability to extreme heat. This is because multiple sources of water ensure a greater amount of water availability for households during times of water scarcity and rationing. In addition, the time spent outside to fetch water for domestic purposes is also likely to be shorter for respondents with more available sources.

Similar to what is observed for Angul, heat vulnerability in Kolkata is lower for respondents that have members with comorbid conditions such as diabetes or hypertension.

Households that avoid the consumption of non-vegetarian foods during the summer months tend to have lower heat vulnerability. This is very much as expected. For instance, meat consumption in the summer season increases the pressure on the digestive system. Meat contains a high amount of fat, proteins, and carbohydrates, which heat up the body during the digestion process of food. Hence, the non-consumption of non-vegetarian food reduces vulnerability to extreme heat.

Occupational (work location) and pre-existing medical (comorbid) conditions, as well as access to resources, were associated with self-reported heat vulnerability, consistent with a study by Tran KV et al. [38].

It should be noted that the approach followed in this study was cross-sectional, which does not allow us to draw definite conclusions about the characteristics associated with heat vulnerability. The data do not allow us to analyse the design and construction of houses, which might have a significant impact on heat vulnerability.

## 5. Conclusions and Policy Recommendations

For the Indian cities of Angul and Kolkata, this paper identified the characteristics of households’ vulnerability to excessive heat and their coping strategies. For Angul, gender, distance (5 km and more) from nearby health facilities, room, temperature, cooling, symptoms, and comorbid factors were the main characteristics, but in the case of Kolkata, occupation (unemployed, agriculture and related), distance (5 km and more), income, room, water, symptoms, comorbid, sleeping (bare floor), and nonvegetarian factors were the main characteristics of households’ vulnerability to extreme heat. This shows that each city has a different set of variables that influences vulnerability, and each factor must be considered while developing a plan to mitigate extreme heat impacts.

A few recommendations follow:An uninterrupted supply of electricity should be provided during summer months, preferably to all households or at least to those households residing in high-risk pockets, i.e., closer to industrial/traffic junctions.When temperatures rise steadily, the use of ACs becomes imminent, especially in regions where the summer temperature can shoot up to 50 °C. Angul is one of the hottest places in the country. In April, the average heat index is appraised at 55.4 °C (131.7 °F). As temperatures rise, the demand for ACs increases. The more the ACs are used, the warmer the ambience becomes. The use of ACs in summer, thus, propels a vicious cycle of global warming. Finally, people who do not have access to cooling appliances should be encouraged to sleep on bare floors (with safety precautions) during extreme heat season. As a short-term solution to counter extreme heat and heatwaves, the state should initiate a ‘Cool Roof’ program in the city.The Angul administration must ensure that all homes, especially in high-heat-risk zones, have access to inexpensive, dependable, sustainable, and modern cooking energy. Awareness should be created among all people (and not just women) in the city regarding the issue and the serious threats that the use of coal poses to their health and well-being, making them more vulnerable to heatwaves and extreme heat.In the case of Kolkata, the study finds that extreme heat is likely to make unemployed people more vulnerable. The government of West Bengal launched ‘Yuvasree’, a financial assistance scheme for the unemployed youth of the state in the year 2013.Efforts must be made to ensure that the urban poor households in Kolkata have access to an adequate supply of quality water throughout the summer.The state should devise policies to reduce meat consumption in Kolkata. A reduction in meat consumption could be beneficial both for climate (through smaller greenhouse gas emissions) and human health. Awareness programs should be conducted to encourage the local population to avoid red meat during the summer months.Providing adequate care to comorbid family members entails a high risk of a financial burden on relevant households [31]. Usually, elderly people are devoid of any easy access to healthcare services. Both the Odisha and West Bengal governments should offer free healthcare to senior citizens at designated public healthcare centres in Angul and Kolkata, respectively.

## Figures and Tables

**Table 1 ijerph-19-15334-t001:** Description of independent variables for Angul and Kolkata.

	Variable	Measurement	Likely Impact	
Angul	Gender of the respondent (Gender)	Gender of the respondent is defined as a dummy variable. It takes a value 1 if the respondent is female and 0 otherwise.	Negative	[24,25,26]
	Distance of the nearest primary healthcare centre from the place of stay (Distance)	Distance of the nearest primary healthcare centre from the place of stay is measured as an ordinal categorical variable: 1 = less than 1 km; 2 = between 1 km and 5 km; 3 = more than 5 km.	Positive	
	Household size (Size)	It is measured as the absolute number of family members in a house.	Positive	
	Number of rooms in the house (Room).	It is measured as the absolute number of rooms in a house.	Negative	[27]
	Perceived change in temperature	Household’s perception of changes in the level of temperature is defined as a categorical variable, with ‘1′ increasing slightly and ‘2′ increasing drastically.	Negative	
	Use of air coolers or air conditioners at the workplace (Cooling)	It is measured as a dummy variable. It takes the value ‘1′ if there is the use of air-coolers or air-conditioners at a workplace and ‘0′ otherwise.	Negative	
	Mild symptoms (Symptoms)	It is measured as a dummy variable. It takes the value ‘1′ if the household head has experienced mild symptoms of high ambient heat such as headache, dizziness, weakness, and muscle pain during summers and ‘0′ otherwise.	Negative	[28]
	Comorbid conditions in the household members (Comorbid)	It is measured as a dummy variable. It takes the value ‘1′ if any member of the household has diabetes and/or hypertension and ‘0′ otherwise.	Unknown	[28]
Kolkata	Occupation of the household head (Occupation)	Occupation of the household head is defined as a categorical variable: 1 = professional/semi-professional; 2 = clerical; 3 = skilled/semi-skilled; 4 = unskilled; 5 = unemployed; 6 = self-employed/business; 7 = agriculture/allied.	Unknown	
	Distance of the nearest primary healthcare centre from the place of residence (Distance)	Distance of the nearest primary healthcare centre from the place of residence is measured as an ordinal categorical variable: 1 = less than 1 km; 2 = between 1 km and 5 km; 3 = more than 5 km.	Positive	
	Average income of the household in summer (Income)	It is measured as the natural logarithm of the average household income during summer months.	Negative	
	Number of rooms in the house (Room).	It is measured as the absolute number of rooms in a house.	Negative	[27]
	Sources of water (Water)	It is measured as the absolute number of sources of water accessible to households.	Negative	
	Mild symptoms (Symptoms)	It is measured as a dummy variable. It takes the value ‘1′ if the household head has experienced mild symptoms of high ambient heat such as headache, dizziness, weakness, and muscle pain during summers and ‘0′ otherwise.	Positive	
	Comorbid conditions in the household members (Comorbid)	It is measured as a dummy variable. It takes the value ‘1′ if any member of the household has diabetes and/or hypertension and ‘0′ otherwise.	Positive	[28]
	Place of sleeping (Sleeping)	Place of sleeping is measured as a categorical variable: 0 = bed, 1 = bare floor, 2 = mattress floor, and 3 = terrace.	Unknown	
	Intake of nonvegetarian foods (Nonveg)	Intake of nonvegetarian foods is defined as a dummy variable. It takes the value ‘1′ if the household avoids the intake of nonvegetarian foods during extreme summers and ‘0′ otherwise.	Negative	
	Changes in the food consumption amount (Food)	Change in the amount of food consumption is defined as a dummy variable. It takes the value ‘1′ if the household has reduced the amount of food consumption during summers and ‘0′ otherwise.	Negative	

**Table 2 ijerph-19-15334-t002:** Crosstabulation of HVI with independent variables (Angul and Kolkata).

	Variable	HVI	Low Vulnerability	High Vulnerability	Total	Pearson Chi-Square
Angul	Gender	Male	61 (35.1)	113 (64.9)	174 (100)	10.01 ***
		Female	74 (22.0)	262 (78.0)	336 (100)	
	Distance	Less than 1 km	40 (44.4)	50 (55.6)	90 (100)	77.89 ***
		Between 1 km and 5 km	78 (40.8)	113 (59.2)	191 (100)	
		More than 5 km	17 (07.4)	212 (92.6)	229 (100)	
	Temperature	Slightly increased	111 (29.5)	265 (70.5)	376 (100)	6.84 ***
		Drastically increased	24 (17.9)	110 (82.1)	134 (100)	
	Cooling	Using ACs/coolers at workplace	84 (94.4)	5 (5.6)	89 (100)	255.47 ***
		Not using ACs/coolers at workplace	51 (12.1)	370 (87.9)	421 (100)	
	Symptoms	Yes	127 (30.0)	296 (70.0)	423 (100)	16.08 ***
		No	08 (09.2)	79 (90.8)	87 (100)	
	Comorbid	Yes	42 (45.2)	51 (54.8)	93 (100)	20.41 ***
		No	93 (22.3)	324 (77.7)	417 (100)	
Kolkata	Occupation	Professional/Semi-professional	30 (34.1)	58 (65.9)	88 (100)	13.04 **
		Clerical	08 (17.0)	39 (83.0)	47 (100)	
		Skilled/semi-skilled	31 (34.1)	60 (65.9)	91 (100)	
		Unskilled	20 (31.7)	43 (68.3)	63 (100)	
		Unemployed	26 (27.4)	69 (72.6)	95 (100)	
		Self-employed/business	44 (40.7)	64 (59.3)	108 (100)	
		Agriculture and allied	05 (62.5)	03 (37.5)	08 (100)	
	Distance	Less than 1 km	29 (22.3)	101 (77.7)	130 (100)	44.18 ***
		Between 1 km and 5 km	50 (23.8)	160 (76.2)	210 (100)	
		More than 5 km	85 (53.1)	75 (46.9)	160 (100)	
	Symptoms	Yes	72 (42.9)	96 (57.1)	168 (100)	11.61 ***
		No	92 (27.7)	240 (72.3)	332 (100)	
	Comorbid	Yes	47 (43.5)	61 (56.5)	108 (100)	07.18 ***
		No	117 (29.8)	275 (70.2)	392 (100)	
	Sleeping	Bed	116 (28.3)	294 (71.7)	410 (100)	23.66 ***
		Bare Floor	47 (55.3)	38 (44.7)	85 (100)	
		Mattress Floor	1 (20.0)	4 (80.0)	5 (100)	
	Nonveg	Yes	77 (42.3)	105 (57.7)	182 (100)	11.74 ***
		No	87 (27.4)	231 (72.6)	318 (100)	
	Food	Yes	46 (36.8)	79 (63.2)	125 (100)	7.21 *
		No	118 (31.5)	257 (68.5)	375 (100)	

Note: Figures in the parenthesis refer to percentage share in total. * Significant at 10%; ** Significant at 5%; *** Significant at 1%. Source: Primary survey.

**Table 3 ijerph-19-15334-t003:** Logistic regression on characteristics of household Vulnerability to extreme HEAT (Angul and Kolkata).

	Variable		Coefficient	Robust SE	z- Statistic	
Angul	Gender		1.155 ***	0.351	3.29	Likelihood Ratio χ^2^ (9): 326.8 *** Log Pseudo Likelihood: −131.34 Pseudo R^2^: 0.55 Pearson χ^2^ (285) ^β^: 296.76 (0.30) ^α^ Hosmer–Lemeshow χ^2^ (8) ^β^: 5.97 (0.65) ^α^ Number of Obsn: 510
	Distance Base: less than 1 km	>1 Km and <5 Km	−0.311	0.405	−0.77	
		5 Km and More	1.650 ***	0.513	3.21	
	Size		0.154	0.105	1.46	
	Room		−0.345 *	0.198	−1.75	
	Temperature		0.881 **	0.433	2.03	
	Cooling		−5.339 ***	0.586	−9.11	
	Symptoms		−1.372 **	0.677	−2.03	
	Comorbid		−1.298 ***	0.384	−3.39	
Kolkata City	Gender	Clerical	0.343	0.518	0.66	Likelihood Ratio χ^2^ (17): 123.2 *** Log Pseudo Likelihood: −254.8 Pseudo R2: 0.195 Pearson χ^2^ (455) ^β^: 463.9 (0.37) ^α^ Hosmer–Lemeshow χ^2^ (8) ^β^: 8.8 (0.36) ^α^ Number of Obsn: 500
		Skilled/semi-skilled	0.195	0.386	0.51	
		Unskilled	0.580	0.447	1.30	
		Unemployed	0.889 **	0.389	2.28	
		Self-employed/business	−0.285	0.369	−0.77	
		Agriculture and related	−1.635 **	0.818	−2.00	
	Distance Base: less than 1 km	>1 km and <5 km	−0.248	0.294	−0.84	
		5 km and more	−1.455 ***	0.312	−4.66	
	Income		0.532 ***	0.152	3.49	
	Room		−0.231 **	0.103	−2.22	
	Water		−0.431 **	0.188	−2.28	
	Symptoms		−0.702 ***	0.257	−2.73	
	Comorbid		−1.075 ***	0.264	−4.06	
	Sleeping Base: Bed	Bare floor	−1.229 ***	0.311	−3.94	
		Mattress floor	0.997	1.366	0.73	
	Nonveg		−0.481 *	0.268	−1.80	
	Food		0.387	0.283	1.37	

Note: * Significant at 10%; ** Significant at 5%; *** Significant at 1%. ^β^ Degrees of freedom for the χ2 statistic; ^α^ indicates the level of significance. Source: Primary survey.

## Data Availability

Survey data are available from the authors however this requires government approval for sharing purposes.

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
