# Peer review of "Characteristics of Households’ Vulnerability to Extreme Heat: An Analytical Cross-Sectional Study from India"

_ijerph, 2022, doi:10.3390/ijerph192215334_

Round 1

Reviewer 1 Report

This paper investigates characteristics of households’ vulnerability to extreme heat by using an analytical cross-sectional study from India. In 2020, a cross-sectional design was applied through 500 households (HHs) each in Angul and Kolkata city. Bi-variate and multi-variable logistic regression analysis were used with Heat Vulnerability Index (HVI) as the outcome variable to identify the determinants of high vulnerability to heat. The results have shown that some common and some different factors are responsible for determining the heat vulnerability of a household across different cities. 

This study provides novel and interesting results. Meanwhile, the authors should consider the following issues:

-There is need for a detailed proofreading of the paper. Currently, it has typing errors and grammatical mistakes. A marked revised paper, which includes some required revisions and comments, is attached below. The addition of the following recent publications referring to development of vulnerability indexes can help to enrich the References part: 

1-"Improved vulnerability index methodology to quantify seismic risk and loss assessment in reinforced concrete buildings". Journal of Earthquake Engineering,.2021. https://doi.org/10.1080/13632469.2021.1911888.

2-"Development of a uniform seismic vulnerability index framework for reinforced concrete building typology". Journal of Building Engineering, 2022. https://doi.org/10.1016/j.jobe.2021.103838

3-"A cross-sectional, randomized cluster sample survey of household vulnerability to extreme heat among slum dwellers in Ahmedabad, India." International Journal of Environmental Research and Public Health. 2013 Jun;10(6):2515-43. https://doi:10.3390/ijerph10062515 

4-"Intra-urban societal vulnerability to extreme heat: the role of heat exposure and the built environment, socioeconomics, and neighborhood stability". Health & Place. 2011 Mar 1;17(2):498-507. https://doi.org/10.1016/j.healthplace.2010.12.005

Author Response

Thanks for your valuable suggestions and comments. We have modified the manuscript.

Reviewer 2 Report

The problems discussed  in the article are important and current, taking into account the increasingly severe effects of climate change including exposure of the population to extreme heat.

Some parts of the article need improvement.

Definition of vulnerability to extreme heat should be explained in the introduction. Factors determining households vulnerability are indicated and explained as well ad other similar terms, but the main term used in the article is not clearly defined. Although the definition is given in point 2.4, the section ‘Materials and methods’ it is not the place for explaining basic terms.

The structure of the whole section 2 is disturbed. The description of methods used in the study should be improved.

Point 2.2 should be developed.

Point 2.3 should be more specific. It is not clear enough if there were 500 respondents total in the study or 1000? What does it mean ‘an almost equal number of households’ in the sentence in lines 97-99?

Division of the point 2.3 for one subsection 2.3.1 is incorrect. Moreover, there is a lack of information about the survey. Wen was it conducted? What was the aim and scope of the survey? What are the main parts of questionnaire?

 Conclusions can contain also recommendations. In my opinion section 5 and 6 could be combined.

Author Response

In the light of your valuable suggestions and comments, we have modified the manuscript.

Reviewer 3 Report

This paper aims to identify the characteristics of households’ vulnerability to extreme heat and their coping practices for people living in the cities of Indian urbanites of Angul and Kolkata in India. The paper is well written, the methodology is sound and the results are well presented.

 I have some comments below to improve the paper:

Methodology- It is not clear why different factors are used to characterise each location?

It appears that use of air conditioners and cooler at workplace are classified as ecological factors. What are the justifications for that? How about presence of fans and coolers in the houses?

The design and construction of houses are not included in the analysis. This will have a significant impact on heat vulnerability and should be included. If not this should be included as the limitation of the study.

In the discussion it is stated that “…………….The finding is paradoxical as numerous studies have shown that women disproportionately suffer the impacts of heat waves (25, 26), disasters (29), and climate change (30, 31) in developing countries, especially in India……”

 Why is it paradoxical? It appears that the findings are similar. This should be revised.

 There are number of typos and formatting errors including units. They should be edited. Some examples as below:

line 80-(temperature ≥45oC),

Line 87-(22,000/km2 as compared to ~200/km2 of Angul).

Line 132 -PHC) from the place of stay is e used to control for the social characteristics

Line 223-on mattress floors (80.0 per cent) havshigh vulnerability to extreme heat, followed by

Line 349-People who experienced mild symptoms of high ambient heat are more likely to stay put

Author Response

In the light of your valuable suggestions and comments, we have modified the manuscript.

Thanks and highly appreciate 

Reviewer 4 Report

Only sugest increase or included citations of the 2022.

Author Response

Thanks and highly appreciate your review.

We have modified the manuscript.
